# Can the Correlation of Periodontopathies with Gastrointestinal Diseases Be Used as Indicators in Severe Colorectal Diseases?

**DOI:** 10.3390/biomedicines11020402

**Published:** 2023-01-29

**Authors:** Lavinia Alina Rat, Andrada Florina Moldovan, Daniela Florina Trifan, Loredana Matiș, Gelu Florin Murvai, Lavinia Maris, Timea Claudia Ghitea, Marius Adrian Maghiar

**Affiliations:** 1Faculty of Medicine and Pharmacy, Medicine Department, University of Oradea, 410068 Oradea, Romania; 2Faculty of Medicine and Pharmacy, Pharmacy Department, University of Oradea, 410068 Oradea, Romania

**Keywords:** periodontopathy, gingival crevicular fluid, inflammation, gastrointestinal disease, quality of life

## Abstract

Gastrointestinal problems are among the most common health problems which can acutely affect the healthy population and chronically involve health risks, seriously affecting the quality of life. Identifying the risk of gastrointestinal diseases in the early phase by indirect methods can increase the healing rate and the quality of life.: The proposal of this study is to verify a correlation between gastrointestinal and periodontal problems and the risk of inflammatory gastrointestinal diseases (IBD). The study was conducted on 123 people who were observed to have gastrointestinal and psychological problems. The participants were divided into three groups, depending on each one’s diagnosis. The control group (CG) was composed of 37 people who did not fit either irritable bowel syndrome (IBS) according to the ROME IV criteria, nor were inflammatory markers positive for IBD. Group 2 (IBS) was composed of 44 participants diagnosed with IBS according to the ROME IV criteria. Group 3 was composed of 42 participants who were diagnosed with IBD. All study participants underwent anthropometric, micro-Ident, and quality of life tests. A directly proportional relationship of the presence of bacteria with IBD patients with the exception of *Capnocytophaga spp.* and *Actinobacillus actinomycetemcomitans* was observed. These two bacteria correlated significantly with IBS. Follow-up of the study participants will help determine whether periodontal disease can be used as an indicator of severe colorectal disease. In addition, this study should be continued especially in the case of IBD more thoroughly to follow and reduce the risk of malignancy.

## 1. Introduction

Colorectal cancer represents one of the greatest risks to health, with a high mortality rate, especially in developed countries, being the third cause of mortality among both men and women [1,2,3]. The number of deaths caused by it reaches 915,880, with more than 230,000 new cases worldwide [2,3]. Long-term survival rates have increased over the past three decades due to more effective diagnostic methods, high accuracy diagnostic tests, and the use of adjuvant therapy, monitoring treatment techniques [4], and advances in the treatment and management of various types of metastatic disease [5,6]. About 89% of patients now survive within the first year after diagnosis and about 65% survive five years and more [3]. In addition to disease-free survival time, quality of life (QoL) became a paramount unitary measurement designated for cancer suffering patients. The quality of life term refers to the summarization of multidimensional concepts, among which are included physical, social functioning, and emotional ones. Furthermore, assessing QOL in cancer patients can improve our understanding of how cancer and therapy influence patients’ lives and how to adapt treatment strategies [7].

Research using the health-related quality of life model [8] has been introduced as a research model for its understanding. Many studies of foreign and domestic types have applied the basal model of this concept. In following studies of elderly patients [9], with coronary heart disease [10], and different types of cancer (e.g., thyroid cancer) [11,12], the main ideas to follow are physiological characteristics, disease awareness, functional status, including symptom status, environmental characteristics, and quality of life. This type of assessment was applied to describe the subjects’ quality of life elements by reporting the relationship between quality of life and these main concepts. Pain affects the quality of life to the point of being unable to carry out daily activities [13,14,15,16].

In terms of looking at the factors that may have an effect on the quality of life of colorectal affectionate patients, additional to social support, self-efficacy, symptom experience, and health promotion behavior and smoking or alcohol consumption have been reported as major variables [8].

Oxidative stress is an aggravating factor in any chronic disease [17], but in gastrointestinal diseases it can have an aggravating impact accelerating the process of malignancy of the disease. It was observed that the levels of protein carbonyl and advanced oxidation protein products (AOPP) were significantly higher in those with colorectal cancer, and the activity of antioxidant enzymes was significantly decreased as well as the serum concentrations of vitamins C and E [18]. Dietary factors have been reported to account for up to 75% of sporadic colorectal cancer, but the mechanisms remain obscure [19]. Epidemiological data suggest that obesity is associated with increased risk of colon cancer for up to 30–70%, especially in men [20].

*Fusobacterium nucleatum*, *Bacteroides forsythus*, and *Treponema denticola* have been identified as potential oncogenic bacteria [21,22,23,24,25]. *Peptostreptococcus* has been associated with endocarditis [26], *Campylobacter rectus* and *Porphyromonas gingivalis* with cytokines and inflammation [27], *Eubacterium nodatum* and *Prevotella intermedia* with descending necrotizing mediastinitis [28], and *Eikenella corrodens* with intra-abdominal infections [29]. *Capnocytophaga spp*. have been correlated with systemic infections [30], and *Actinobacillus actinomycetemcomitans* in coronary heart diseases including cardiomyopathies [31]. These diseases, which can have a negative impact on the body and a course with a high severity, are based on periodontal infection with the presence of the studied bacteria.

This report is an initial report of our investigation into whether or not there is a link between oral bacteria and symptoms of gastrointestinal disease. The objective of this paper is to evaluate a correlation between gastrointestinal problems and the presence of periodontal bacteria, which could be an indicator of the risk of serious inflammatory intestinal diseases, including colorectal neoplasia. The goal is early detection of the stages with very good chances of treatment and survival.

## 2. Materials and Methods

The study was carried out between January 2020 and December 2021, with patients being presented at the Echo Laboratories private medical office with gastrointestinal symptoms. From here, the selected patients were sent to different specialists for specific analyses (gastroenterology, dentistry, psychology).

### 2.1. Body Analysis of Patients

The dental support and sampling were provided by Dr. Călinescu Delia’s private medical office of dentistry and the paraclinical and clinical analyses were performed in Oradea, in a private medical office specialized in nutrition “Echo Laboratoare”.

#### Anthropometric Tests

There was conducted a cross-sectional study on the metabolic syndrome (MS). These patients were enrolled in a cohort study, to obtain the clinical characteristics of MS patients. In the study conducted, the patients were given a personalized diet, a healthy and clinical one (hypocaloric intake; thus, macronutrients in the percentage of 40–55% carbohydrates, 15–20% lipids, and 25–35% proteins, with a maximum reduced caloric intake of 200 kcal). The personalization of the clinical diet was recommended following venous/capillary blood testing of a specific food allergic reaction of immunoglobulin G (IgG) type 3 and 4 to a number of 90 foods specific to the local area and cuisine. The foods to which a specific IgG reaction was recorded were excluded from the diet for a period of 3 months, only occasionally reintroduced, until the research period ended (one year). Clinical evaluation was performed with the Tanita MC780MA, a bioelectrical impedance body analyzer (BIA) (Tokyo, Japan) [32]. The results were evaluated using medical software GMON 3.4.1 (Chemnitz, Germany). BIA body analyzers are highly accurate devices that are WPHNA (World Public Health Nutrition Association) approved and have been used to determine body composition. The margin of error was 0.1 kg. The therapeutic diet tracked changes measured with the noninvasive Cnoga MTX medical device (Or-Akiva, Israel), which helped track changes in general clinical parameters by checking blood pressure, oxygen saturation, and blood pH. Patients were evaluated on an empty stomach in the morning.

The variations in the three independent groups, were followed according to sex, rural/urban environment, age, clinical parameters, such as BAM, weight status, fat mass, visceral fat, hydration status, extracellular water/total body water (ECW/TBW), sarcopenia index, angle of phase, basal metabolic rate (BMR), and pH, followed by MS.

At the beginning and at the end of the study period, the participants completed evaluation forms.

### 2.2. Collection of Gingival Crevicular Fluid

The patients who had inflamed or bleeding gums (periodontopathies) presented themselves in the mentioned dental office, where the dentist assessed the patients’ periodontal health. The gingival crevicular fluid was collected by the stomatology doctor from the periodontal sac, not from the dental plaque, precisely to avoid confusion between the presence of plaque-forming bacteria and periodontal bacteria from the periodontal pulp. An exclusion criterion was “advanced periodontal disease.

### 2.3. Micro-IDent Test

Deoxyribonucleic acid (DNA) testing of gingival crevicular fluid for the presence of bacteria.

Gingival fluid was collected with dry paper tips if the pocket depth > 4 mm with BOP (despite excellent oral hygiene). The dry tip of the paper is held deep in the pocket for 15 s until it completely absorbs the gingival crevicular fluid. The paper tip is placed in the kit box, sealed, and stored at 2–8 °C until processing.

DNA extraction and manipulation from dried paper blots. 

An extraction kit of bacterial DNA (HAIN Lifescience) from dry paper spots was used for extraction according to the manufacturer’s instructions. Briefly, in a flask containing each periodontal sample, 100 μL of lysis solution was added and vigorously mixed for 10 s to elute the bacterial cells from the paper points. At 95 °C on the PeQLab heating block (Biotechnology, GmbH, Germany), it was incubated for 5 min with a strong vortex every 2 min, after which the mixture was allowed to cool to room temperature. Then, a volume of 100 mL of neutralization buffer was added, and the solution was centrifuged in a Mikro 200 microcentrifuge (Centrifugen, Hettich, Germany) for 10 min at 14,000 rpm. The volume used as a template for the subsequent amplification step was 5.0 μL of supernatant. This was stored at low temperatures, to be exact at −20 °C pending hybridization.

The amplification was performed by polymerase chain reaction (PCR), containing 50 μL reaction volume of 5.0 μL of template DNA and 45 μL of reaction mixture containing 35 μL of primer-nucleotide-PNM (Micro-IDent^®^plus) mixture, 5.0 μL of 10× PCR buffer, 5.0 μL of 2.5 mM MgCl_2_, and 0.2 U Taq at the hot start (Qiagen, GmbH, Germany). PCR cycling was performed in a GTQ-Cycler 96 thermal cycler (HAIN Lifescience, Nehren, Germany). An initial stage of denaturation at 95 °C for 5 min included the cycling conditions, 10 cycles at 95 °C for 30 s and at 60 °C for 2 min, 20 cycles at 95 °C for 10 s, at 55 °C for 30 s, and at 72 °C for 30 s, and a final step extension to 72 °C for 10 min. The test sample included the negative control. The negative control was formed 5.0 μL of sterile PCR water, each added to 45 μL of the reaction mixture. Subsequently, the reverse hybridization was performed according to the Micro-IDent^®^plus test (HAIN Lifescience GmbH, Nehren, Germany). Biotinylated amplicon was denatured and incubated in the Twincubator^®^ (HAIN Lifescience) at 45 °C with a hybridization buffer and coated strips with two control lines (Conjugates and amplification) with six (Micro-IDent^®^plus) species-specific samples. A very specific washing step which removes any nonspecific bound DNA ensued. After this, the PCR products were bound to their respective probe complements. The alkaline phosphatase-conjugated was added to streptavidin. The samples were washed, and the hybridization products were visualized by adding an alkaline phosphatase substrate. A total of 11 selected species of periodontopathogenic bacteria can be identified using this test [33].

### 2.4. Statistical Analysis

The exit rate was reported of the individuals in the study as 0.00%; therefore, the statistical significance Asymp was obtained using the Chi-Square test (*p* < 0.05). SPSS software (version 20) was used to perform statistical analysis. All means, ranges, frequency tests of statistical significance, and standard deviations were summarized using Student’s t test. The coefficient of the ANOVA statistical test “F” is based on 3 or more mismatched pairs or groups, and “t” represents the coordinate for the Student’s *t*-test statistic for 2 independent groups in this case.

Group distributions were similar to normal, using assumptions involving numerical data. The Bravais–Pearson correlation coefficient was used to calculate an independent indicator of the two variables. Statistical significance of the analysis of variance (ANOVA) is recorded at a *p*-value less than 0.05, and a *p*-value of 0.01 indicated a high level of significance. Differences between groups were followed by Bonferroni post hoc analysis as subgroup analyses.

### 2.5. Ethical Considerations

The current study was completed with the approval of the Research Ethics Commission of the Faculty of Medicine and Pharmacy, University of Oradea, Romania no. (12/01.04.2019), by the approval of ECHO Laboratories (no. 09/01.04.2017), just as specified by the guidelines of the Declaration of Helsinki (Ethical principles for medical research involving human subjects) [34]. In the research, all patients incorporated agreed for their data to be processed and signed an informant consent form before their inclusion in the study, according with current legislation.

### 2.6. Study Groups

The study involved 123 people with MS who were observed to have gastrointestinal and psychological problems. The Micro-Ident test, anthropometric evaluation, and quality of life tests was performed on all participants. The participants were divided into 3 groups, depending on each one’s diagnosis. The control group (CG) was composed of 37 people who did not fit either irritable bowel syndrome (IBS) according to the ROME IV criteria, nor were inflammatory markers positive for IBD. Group 2 (IBS) was composed of 44 participants diagnosed with IBS according to the ROME IV criteria. Group 3 was composed of 42 participants who were diagnosed with IBD.

There were no significant differences (X^2^ = 0.634, *p* = 0.728) in the health status or quality of life of the patients in the 3 groups.

### 2.7. Exclusion Criteria

Exclusions were advanced periodontal disease or colorectal neoplasia and patients on psychiatric antidepressant treatments.

## 3. Results

The initial data show that the cohort consists of 65 men and 58 women (X^2^ = 0.398, *p* = 0.528), with 52 from urban environments and 71 from rural environments, and having an average age of 40.27 years, standard deviation (SD) 15.35 years, as shown in Table 1.

Baseline anthropometric parameters were thus cohort mean body mass index (BMI) 30.85 (SD 7.69) t = 0.216 *p* = 0.806, fat mass 31.16 (SD 8.70) t = 39.690 *p* = 0.001, visceral fat = 7.97 *p* = 0.001 (SD 5.60). The statistical description of the baseline data for each group is shown in Table 2.

Insignificant differences in BMI were recorded between the CG and the IBS, between the CG and the IBD, but also between the IBS and the IBD. In the case of fat mass and visceral fat, insignificant differences were registered between the CG and the IBD, and between the IBS and the IBD. Insignificant differences were also observed between the CG and the IBS.

The age distribution according to the study group is shown in Figure 1. Thus, peri-odontal infections tested by Micro-IDent^®^plus tests were confirmed even at young ages (mean 37.86 ± 13.21 years) in CG, and 40.73 ± 16.22 in IBS group, and 41.90 ± 16.23 in IBD group.

### 3.1. Bacterial Evaluation of Patients (at Risk of Colorectal Cancer)

Briefly, 11 specific batteries from gingival crevicular fluid were evaluated, namely Peptostreptococcus micros, Campylobacter rectus, Eubacterium nodatum, Eikenella corrodens, *Capnocytophaga* spp., Actinobacillus actinomycetemcomitans, Prevotella intermedia, Porphyromonas gingivalis, Bacteroides forsythus, and Treponema denticola. These bacteria have been correlated with periodontal disease, but also with inflammation [27], cancers [35], with prostate diseases [36], and even with Alzheimer’s [37]. All these diseases with a high severity are based on the periodontal infection with the presence of the bacteria studied.

From a bacteriological point of view, 11 bacteria from the gingival crevicular fluid were tested, and the presence of *Fusobacterium nucleatum* was followed (Figure 2). It was observed to be most frequently present in the gingival crevicular fluid in 100.0% of the total study subjects.

The frequency of periodontal infection, presented in Figure 3, emphasizes that 39 patients presented an infection with six bacteria out of the 11.

The distribution of bacteria presence in the three research groups is described in Table 3 and in Figure 4. *Prevotella intermedia* is present in all of the patients in the IBD groups and in 0 patients in the CG and IBS groups. In the CG group, the presence of eight different bacteria can be observed, in IBS, nine bacteria are present, and in IBD, all 11 bacteria are present in the gingival crevicular fluid. Although all the bacteria in the study are present in IBD, they are the most common in IBS (according to Figure 4).

### 3.2. Evaluation of Gastrointestinal Problems

Evaluating gastrointestinal problems, nausea/vomiting, flatulence, diarrhea, constipation, and malnutrition were observed in the three study groups, shown in Table 4.

The gastrointestinal and psychological parameters were estimated by the patients completing an assessment form at the beginning of the study period. The absence/presence of symptoms were noted on the sheets with “0” or “1”.

Statistically significant differences were recorded in the case of nausea between the research groups (*p* = 0.012), the largest difference being recorded between CG and IBS (*p* = 0.015). Significant differences were also recorded between CG and IBD (*p* = 0.05), but between IBS and IBD the differences are insignificant. An increase in the incidence of flatulence and diarrhea was observed, and the differences are statistically significant (*p* = 0.021 for diarrhea and *p* = 0.001 for diarrhea). Regarding constipation, it differs insignificantly in each group (*p* = 0.092), the most being present in IBS. Malnutrition differs between the three groups significantly (*p* = 0.011), and insignificant differences were recorded in two research groups (CG and IBS) but still insignificant between IBS and IBD (*p* = 1.000). There was no significant difference in malnutrition between the three groups. Constipation was the most present in the IBS group, but due to non-significant differences, it cannot be related to the presence of bacteria. Severe malnutrition was observed in only two patients, in group 3 (with periodontal bacterial infection). Moderate malnutrition was observed especially in the IBD group, which is also shown by the statistical significance. 

### 3.3. Evaluation of Psychological Factors (Stress, Anxiety, Depression)

The assessment of psychological risk factors was based on the completion of self-report questionnaires, whether the participants feel more stressed, anxious, or depressed, as described in Table 5. These assessments were based 100% on the individual’s perception, which was later correlated with the completed questionnaires in the presence of the psychologist, in order to monitor the quality of life.

### 3.4. General Risk Factors

Among the general risk factors, smoking and alcohol consumption were tracked. The majority of alcohol users are in the IBS group, smoking is present in IBD, and there are statistically insignificant differences between the three groups. The description of the risk factors in the three groups can be found in Table 6.

### 3.5. Correlations

The directly proportional relationship between the presence of bacteria and the research groups shows a strong link with statistical significance, as can be found in Table 7. Thus, the number of bacteria and infections is higher in the IBD group. In the case of *Capnocytophaga spp*. and *Actinobacillus actinomycetemcomitans* bacteria, a negative value of the Pearson coefficient can be observed, which indicates an inversely proportional but statistically significant relationship with IBD.

### 3.6. Quality of Life Study

Quality of life was assessed with the help of three European Organisation for Research and Treatment of Cancer colorectal quality of life (EORTC QLQ tests–CR29), visual analog scale (VAS), and followed the specific World Cancer Research Fund (WCRF) and the American Institute for Cancer Research (AICR) recommendations.

The EORTC QLQ–CR29 is a quality of life questionnaire, and it looks at it through the lens of gastrointestinal problems. Thus, it evaluates the degree of impairment of the quality of life (89–108 points: severe impairment, 48–88 bridge: moderate impairment, and 27–47 points: minor impairment).

The VAS quality of life test tracks pain and tolerability. The number of points collected represents different degrees of pain, where tolerance can be very different in each individual.

So, a minor but existing pain (0–3 points) reflects the first decline in quality of life, severe pain, but not affecting daily activities (4–7 points), and very serious, severe pain, including daily activities (8–10 points) shows severe impairment of quality of life.

In the case of WCRF/AICR, special recommendations and their compliance can be followed for those with gastrointestinal diseases. The interpretation of the results of the 10 questions is divided as follows: 0–2 does not comply with the recommendations, 3–5 partial compliance, 6–8—good compliance, 9–10 strict compliance.

EORTC QLQ–CR29 is a quality of life questionnaire, and it looks at it through the prism of gastrointestinal problems. Thus, it evaluates the degree of impairment of the quality of life (89–108 points: severe impairment, 48–88 points: moderate impairment, and 27–47 points: minor impairment).

The VAS quality-of-life test measures pain and tolerability. The number of points collected represents different degrees of pain, where the tolerance can be very different for each individual.

So a minor but existing pain (0–3 points), reflects the first decline in the quality of life, severe pain, but not affecting daily activities (4–7 points), and very serious, severe pain, affecting even daily activities (8–10 points) shows severe impairment of quality of life.

In the case of WCRF/AICR, you can follow special recommendations and their compliance for those with gastrointestinal diseases. The interpretation of the results from the 10 questions is divided as follows: 0–2 does not comply with the recommendations, 3–5 partial compliance, 6–8 good compliance, 9–10 rigorous compliance. The statistical description of the three tests is presented in Table 8 and Figure 5.

## 4. Discussions

Obesity as a disease of the 21st century is one of the most common [38]. The victims of the modern food industry are faced not only with obesity, but also with metabolic and non-metabolic diseases associated with it [39,40]. A proinflammatory process that characterizes obesity [41,42] is the basis of Th2 [43] or IgG [44]-type reactions, mediated mostly by food. Gastrointestinal damage, mediated by lifestyle imbalance, particularly an excessive intake of refined carbohydrates [45], leads to a significant increase in hypertension diseases [46], dyslipidemia [47], and cardio-metabolic diseases [48]. In this study, it was observed that the patients had a high BMI of 30.85 on average, which is obesity grade I. Among the three groups, the highest value of BMI was recorded in the IBS group, but with small, statistically insignificant differences. In addition, in those with IBD, malnutrition is present significantly more, which can explain both the BMI and the fat mass or visceral fat lower in the IBD group. Visceral fat is correlated not only with metabolic disease [49], but it is considered an aggravating factor in the neoplastic diseases [50]. Patients in the current study presented with visceral fat at an average level of 7.97, but with a large standard deviation, which showed that in each group there are people with high visceral fat, which is considered at risk for the body.

After assessing the psychological factors, 87.0% were assessed with stress, all presented with anxious symptoms and only 30.1% with mild depression. Specialized studies show a direct link between gastrointestinal diseases and stress [51]. Anxiety has been reported among between 50% and 90% of patients [52]. Dysthymia or depression has been followed in the con-text of gastrointestinal diseases by Mudyanadzo [53]. He observed an amplification of inflammatory bowel disease when depression was also present.

The presence of bacteria in the gingival crevicular fluid has been correlated with a pro-inflammatory process [54], but studies also emphasize a connection with colorectal cancer [22,55,56,57], in neurodegenerative diseases [58], with cardiomyopathological risk [59], with arthritis [60], prostatic disease [61], or with proinflammatory process [27]. In the current study, the gingival crevicular fluid analyzes showed a high incidence of *Fusobacterium nucleatum* in patients with gastrointestinal problems (100.0%).

Gupta in 2013 drew a parallel between the presence of bacterial infection and the secretion response of inflammatory mediators, as a trigger for the progression of periodontal [62] and intestinal inflammatory diseases [63]. Another 2021 study published in Turkey [64] followed children in the evolution of chronic inflammatory bowel disease (IBS) and cytokine-mediated inflammation (interleukin (IL)-1β, IL-12, IL-21, IL-22, IL -23, IL-17A, IL-17F). In medical practice, it is difficult to distinguish IBS from IBD from a symptomatic point of view. The analyses of specific inflammatory agents in the case of IBD make the difference, and the permanent damage to the intestinal mucosa [65]. Active IBS has been found to increase gingival inflammation. This study shows the reciprocal activation of the pro-inflammatory process, either triggered by periodontal or intestinal bacteria, but the link is direct, and if treatment or prevention measures are not taken, they can lead to the development of serious chronic diseases.

Castellarin, in a 2012 study, verified the presence of the bacterium *Fusobacterium nucleatum* in patients with colorectal carcinoma [66]. Following the PCR analyses, it was found that this bacterium was present in 99.3% of the patients. In the current study, the IBD group recorded the highest number of periodontal bacteria present. Significantly more pronounced gastrointestinal symptoms were registered in IBD. This process represents a major risk for health, a risk that includes both an inflammatory effect on the intestinal mucosa and a risk of colorectal cancer.

A study published by Giannopoulou concluded that increased production of inflammatory cytokines (IL-4, IL-6, and IL-8) observed in the study, in the presence of smoking, may have clinical consequences. In 2017, Çalapkorur considered the pro-inflammatory process mediated by periodontal bacteria in the gingival crevicular fluid. The correlations of the research groups from the current study (IBS and IBD group) with the presence of bacteria is in accordance with specialized studies in the current literature.

For reducing inflammation, treating periodontal infection and gastrointestinal symptoms is important, but perhaps even more important is improving the quality of life. It was found that the presence of gingival inflammation, largely asymptomatic, does not significantly affect the quality of life. Instead, gastrointestinal problems [67] and psychological ones do [68,69]. The duration and intensity of pain especially affect the quality of life of IBS patients, as Mönnikes observed [70]. The quality-of-life tests used are particularly recommended for these pathologies. Our results show that the most affected in terms of quality of life are those from the FG group. Compliance with the recommendations to maintain an ideal body weight or to perform a regular physical activity and to implement a healthy eating style has a major impact in gastrointestinal diseases. Failure to comply with these, together with general risk factors, lead to health impairment, i.e., intestinal health imbalance. Therefore, in patients with specific gastrointestinal symptoms and an altered psychological state, lifestyle changes are indicated.

One of the study’s limitations was the lack of dietary history. At the same time, another pro-inflammatory process at the intestinal level, such as food intolerances, can be the trigger of intestinal inflammation. The presence of chronic gastrointestinal infections with resistant bacteria can also cause intestinal inflammation, as well as the disruption of the microbiota that play an important role in the secretion of anti-inflammatory intestinal substances. Early interventions, by changing the diet, regulating the intestinal microflora (probiotic therapy), and managing stress, as well as reducing risk factors, can contribute to a significant reduction in the risk of developing colorectal cancer and contribute to an increase in the quality of life. 

## 5. Conclusions

A higher BMI was recorded for the IBS group, but the increase was not statistically significant. This may suggest a more complex link with periodontal bacteria. The highest fat mass was observed in the IBS group, which can be explained by the proinflammatory process of the digestive tract and is a high risk of colorectal cancer in IBD. Visceral fat was the highest in the IBS group, which can be explained by the pro-inflammatory process, but there was no permanent damage to the intestinal mucosa.

Among the 11 bacteria present in the gingival crevicular fluid detectable by the micro-IDent test, *Fusobacterium nucleatum* was detected most often. In the IBS group, the presence of more periodontal bacteria was recorded. The most affected quality of life, according to the EORTC QLQ–CR29 test, was in the IBS group (*p* = 0.023).

Pain present but not affecting daily activities, assessed by the VAS test, was observed in all three groups. In the case of the WCRF/AICR test, in terms of compliance with specific recommendations, a partial compliance with the recommendations was observed, with the groups with the highest compliance being the IBD group. Failure to comply with specific recommendations can lead to worsening health and affect quality of life.

Several periodontal bacteria have been identified as potential oncogenic bacteria [21,22,23,24,25]. This is an initial report of our investigation to determine whether or not there is a link between the presence of bacteria in gingival crevicular fluid and symptoms of gastrointestinal disease to determine whether it can be exploited for the early diagnosis of colorectal cancer or preventing the development of colorectal cancer in patients at risk. Colonic irritation and inflammation were observed in all three groups. Follow-up of study participants will help determine whether periodontal diseases can be used as indicators of severe colorectal disease.

## Figures and Tables

**Figure 1 biomedicines-11-00402-f001:**
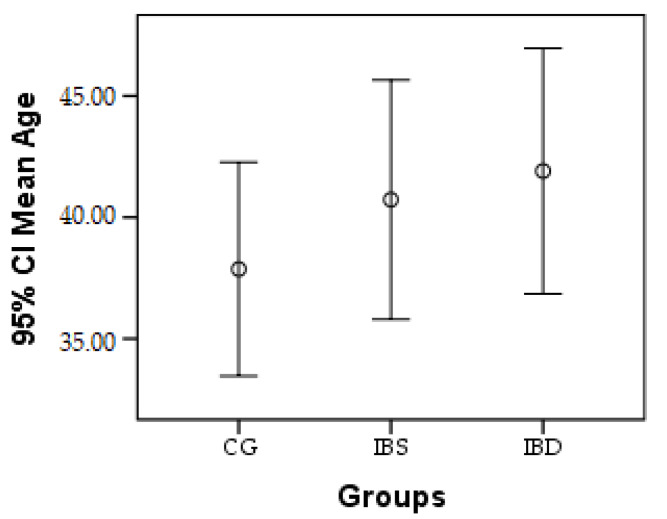
Age distribution by study group, where, dots represent the mean age in each group. The large standard deviation shows a variety of age in all 3 research groups.

**Figure 2 biomedicines-11-00402-f002:**
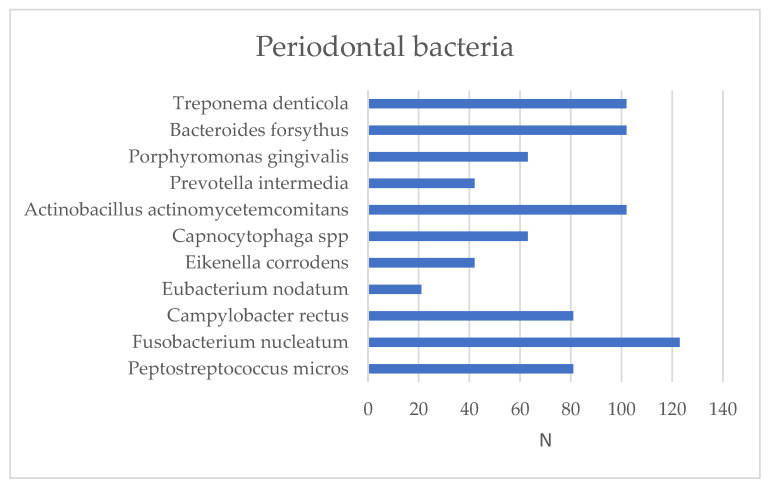
Bacteria present in the gingival crevicular fluid, where N is the number of patients.

**Figure 3 biomedicines-11-00402-f003:**
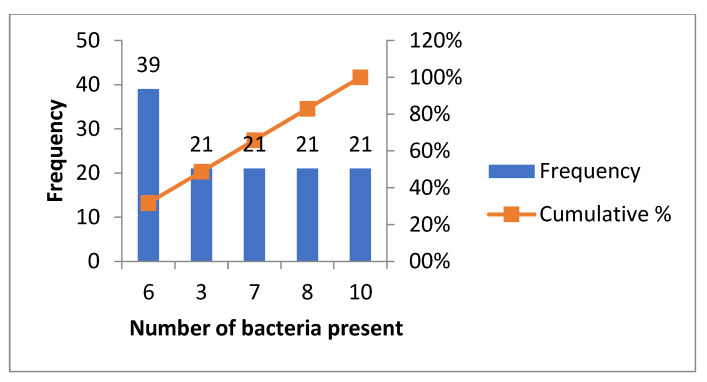
Graphical presentation with the histogram technique, of the number of patients in whom 3 or more bacteria were detected.

**Figure 4 biomedicines-11-00402-f004:**
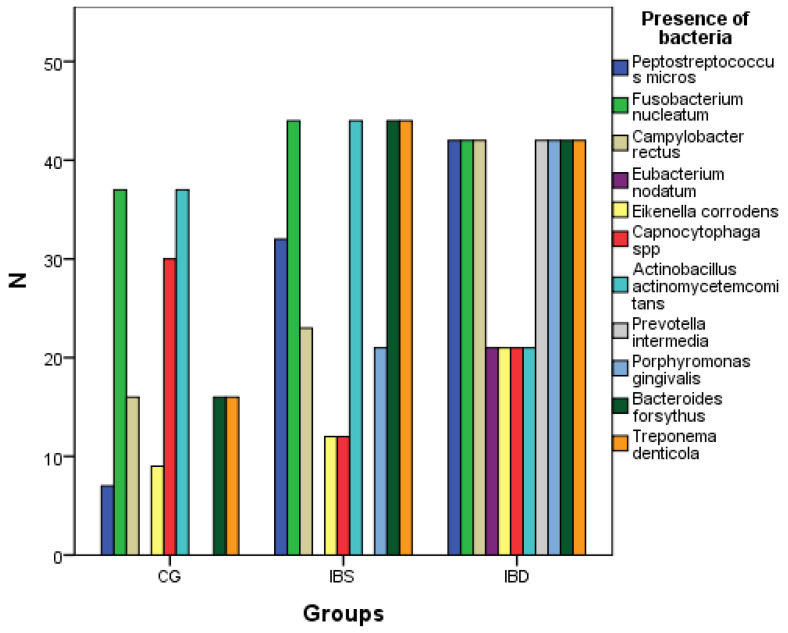
Graphical presentation of the number of patients in the three research groups.

**Figure 5 biomedicines-11-00402-f005:**
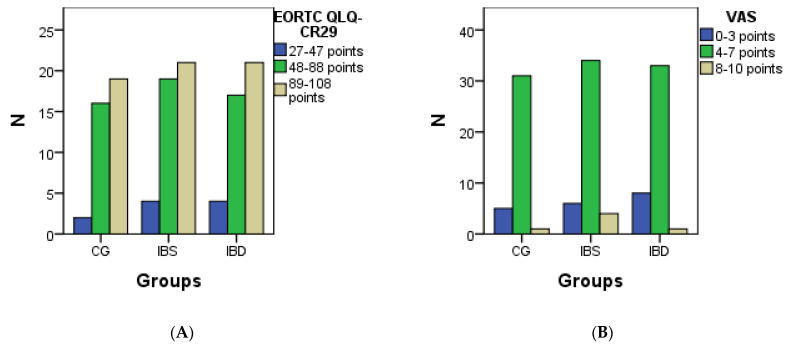
Graphic presentation of the quality of life questionnaires, respectively EORTC QLQ–CR29 (**A**), VAS (**B**), and WCRF/AICR (**C**).

**Table 1 biomedicines-11-00402-t001:** Demographic description of patients.

Parameters		Groups						Total	
		CG		IBS		IBD			
		N	%	N	%	N	%	N	%
Urban/rural	urban	13	35.1	21	47.7	18	42.9	52	42.3
	rural	24	64.9	23	52.3	24	57.1	71	57.7
Gender	male	21	56.8	24	54.5	20	47.6	65	52.8
	female	16	43.2	20	45.5	22	52.4	58	47.2

CG = Control group, IBS = Group with IBS, IBD = Group with IBD, N = number of patients.

**Table 2 biomedicines-11-00402-t002:** Statistical description of baseline data.

Parameters	Groups						Total	
	CG		IBS		IBD			
	Mean	SD	Mean	SD	Mean	SD	Mean	SD
Age	37.86	13.21	40.73	16.22	41.90	16.23	40.27	15.35
BMI	30.13	8.74	31.65	8.10	30.64	6.23	30.85	7.69
Fat mass	30.87	9.37	31.34	8.90	31.22	8.08	31.16	8.71
Visceral fat	6.92	5.87	8.70	5.48	8.12	5.50	7.97	5.61

CG = Control group, IBS = Group with IBS, IBD = Group with IBD, SD = Standard deviation.

**Table 3 biomedicines-11-00402-t003:** Descriptive presentation of the presence of bacteria in the three research groups.

Bacteria		Groups						Total	
		CG		IBS		IBD			
		N	%	N	%	N	%	N	%
*Peptostreptococcus micros*	absent	30	81.1	12	27.3	0	0.0	42	34.1
	present	7	18.9	32	72.7	42	100.0	81	65.9
*Fusobacterium nucleatum*	absent	0	0.0	0	0.0	0	0.0	0	0.0
	present	37	100.0	44	100.0	42	100.0	123	100.0
*Campylobacter rectus*	absent	21	56.8	21	47.7	0	0.0	42	34.1
	present	16	43.2	23	52.3	42	100.0	81	65.9
*Eubacterium nodatum*	absent	37	100.0	44	100.0	21	50.0	102	82.9
	present	0	0.0	0	0.0	21	50.0	21	17.1
*Eikenella corrodens*	absent	28	75.7	32	72.7	21	50.0	81	65.9
	present	9	24.3	12	27.3	21	50.0	42	34.1
*Capnocytophaga* spp.	absent	7	18.9	32	72.7	21	50.0	60	48.8
	present	30	81.1	12	27.3	21	50.0	63	51.2
*Actinobacillus actinomycetemcomitans*	absent	0	0.0	0	0.0	21	50.0	21	17.1
	present	37	100.0	44	100.0	21	50.0	102	82.9
*Prevotella intermedia*	absent	37	100.0	44	100.0	0	0.0	81	65.9
	present	0	0.0	0	0.0	42	100.0	42	34.1
*Porphyromonas gingivalis*	absent	37	100.0	23	52.3	0	0.0	60	48.8
	present	0	0.0	21	47.7	42	100.0	63	51.2
*Bacteroides forsythus*	absent	21	56.8	0	0.0	0	0.0	21	17.1
	present	16	43.2	44	100.0	42	100.0	102	82.9
*Treponema denticola*	absent	21	56.8	0	0.0	0	0.0	21	17.1
	present	16	43.2	44	100.0	42	100.0	102	82.9

CG = Control group, IBS = Group with IBS, IBD = Group with IBD, N = number of patients.

**Table 4 biomedicines-11-00402-t004:** Descriptive presentation of the parameters of gastrointestinal disorders in the three research groups.

Parameters		Groups						Total	
		CG		IBS		IBD			
		N	%	N	%	N	%	N	%
Nausea/vomiting	absent	29	78.4	21	47.7	22	52.4	72	58.5
	present	8	21.6	23	52.3	20	47.6	51	41.5
Flatulence	absent	26	70.3	19	43.2	18	42.9	63	51.2
	present	11	29.7	25	56.8	24	57.1	60	48.8
Diarrhea	absent	24	64.9	9	20.5	15	35.7	48	39.0
	present	13	35.1	35	79.5	27	64.3	75	61.0
Constipation	absent	22	59.5	16	36.4	17	40.8	55	44.7
	present	15	40.5	28	63.6	25	59.5	68	55.3
Malnutrition	absent	20	54.1	11	25.0	7	16.7	38	30.9
	moderate	15	40.5	33	75.0	35	83.3	83	67.5
	severe	2	5.4	0	0.0	0	0.0	2	1.6

CG = Control group, IBS = Group with IBS, IBD = Group with IBD, N = number of patients.

**Table 5 biomedicines-11-00402-t005:** Psychological risk factors.

Parameters		Groups						Total	
		CG		IBS		IBD			
		N	%	N	%	N	%	N	%
Stress	absent	6	16.2	4	9.1	6	14.3	16	13.0
	present	31	83.8	40	90.9	36	85.7	107	87.0
Anxiety	absent	0	0.0	0	0.0	0	0.0	0	0.0
	present	37	100.0	44	100.0	42	100.0	123	100.0
Depression	absent	28	75.7	28	63.6	30	71.4	86	69.9
	present	9	24.3	16	36.4	12	28.6	37	30.1

CG = Control group, IBS = Group with IBS, IBD = Group with IBD, N = number of patients.

**Table 6 biomedicines-11-00402-t006:** Description of risk factors in the three groups.

Parameters		Groups						Total	
		CG		IBS		IBD			
		N	%	N	%	N	%	N	%
Alcohol	absent	32	86.5	37	84.1	37	88.1	106	86.2
	present	5	13.5	7	15.9	5	11.9	17	13.8
Smoking	absent	9	24.3	16	36.4	12	28.6	37	30.1
	present	28	75.7	28	63.6	30	71.4	86	69.9

CG = Control group, IBS = Group with IBS, IBD = Group with IBD, N = number of patients.

**Table 7 biomedicines-11-00402-t007:** Pearson correlation regarding the relationship between the presence of bacteria and research groups.

Pearson Correlation		Groups
Peptostreptococcus micros	r	0.679 **
	p	0.001
Campylobacter rectus	r	0.486 **
	p	0.001
Eubacterium nodatum	r	0.544 **
	p	0.001
Eikenella corrodens	r	0.220 *
	p	0.014
Capnocytophaga spp	r	−0.235 **
	p	0.009
Actinobacillus actinomycetemcomitans	r	−0.544 **
	p	0.001
Prevotella intermedia	r	0.863 **
	p	0,001
Porphyromonas gingivalis	r	0.801 **
	p	0.001
Bacteroides forsythus	r	0.590 **
	p	0.001
Treponema denticola	r	0.590 **
	p	0.001
	N	123

r = Pearson coefficient, p = statistical significance, N = number of patients, ** correlation is significant at the 0.01 level (2-tailed), * correlation is significant at the 0.05 level (2-tailed).

**Table 8 biomedicines-11-00402-t008:** Descriptive statistics of the 3 quality of life tests.

Parameters	Groups												TOTAL	
	CG				IBS				IBD					
	Mean	SD	Min	Max	Mean	SD	Min	Max	Mean	SD	Min	Max	Mean	SD
EORTC QLQ CR29	41.58	2.85	34.20	46.10	41.87	2.95	34.40	48.40	40.93	4.13	30.30	47.40	41.46	3.37
VAS	5.03	1.28	2.00	8.00	5.14	1.56	2.00	8.00	5.07	1.52	2.00	9.00	5.08	1.46
WCRF/AICR	3.86	2.21	.00	9.00	3.34	1.80	.00	8.00	4.12	2.17	.00	9.00	3.76	2.07

CG = Control group, IBS = Group with IBS, IBD = Group with IBD, SD = Standard deviation.

## Data Availability

Not applicable.

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
