# Peer review of "Can the Correlation of Periodontopathies with Gastrointestinal Diseases Be Used as Indicators in Severe Colorectal Diseases?"

_biomedicines, 2023, doi:10.3390/biomedicines11020402_

Round 1
Reviewer 1 Report (Previous Reviewer 2)
The quality of the manuscript has been further improved after revision. So, my recommendation is accept.
Author Response
Primarily, we, the authors of the present manuscript wish to thank you for thoughtful commentary you have provided to improve the quality of the paper. We are very grateful for the time and effort you have devoted to this task. Thank you very much for your trust.

Reviewer 2 Report (Previous Reviewer 1)
The manuscript by Lavinia Alina Rat et al. presents data on gastrointestinal disorders and periodontal infection by 11 bacteria groups. The manuscript is of value, but several changes need to be made before it is suitable for publication.
The Introduction states "Colorectal cancer represents one of the greatest risks to health, with a high mortality rate, especially in developed countries, being the second cause of mortality among both men and women [1]." This statement should cite GLOBOCAN 2020 [https://doi.org/10.3322/caac.21660]. It should also be changed to indicate that colorectal cancers are the second leading cause of deaths due to cancer or possibly, as shown in Table 1 in GLOBOCAN 2020, that colon and rectal cancers are the second leading cause of deaths due to cancer.
The Introduction states "The number of deaths caused by it reaches 250,000, with more than 2700 new cases worldwide [2]." Again, GLOBOCAN 2020 should be cited, and Table 1 indicates that the number of deaths in 2020 due to colon and rectal cancers was 915,880.
The Introduction states "The chances of living has increased...." This should be changed to something like "Long-term survival rates have increased...."
The Introduction states "About 80% of patients now, survive within the first year after diagnosis and about 62% survive 5 years and more [6]." This reference is too old, 2002. A more recent reference should be cited for the current 1-year and 5-year survival rates.
The Methods state "There was conducted a cross-sectional study on the metabolic syndrome (MS). These patients were enrolled in a cohort study, to obtain the clinical characteristics of MS patients." Was this study a separate study from the study reported in the current manuscript? If it was a separate study, is the report published?
The Methods state "A total of 11 selected species of periodontopathogenic bacteria can be identified using this test [33]." Either a list of the bacteria should be provided at this point or the reader should be directed to Table 3. For example (see Table 3).
The Methods state "Without significant differences (X2=0.634, p=0.728) the health status of the patients and the quality of life was evaluated, comparing the 3 groups." This should be changed to something like
"There were no significant differences (X2=0.634, p=0.728) in the health status or quality of life of the patients in the 3 groups."
The Methods states
2.7. Inclusion/exclusion criteria
The inclusion criteria were patients with gastrointestinal symptoms and psychological problems.
This sentence is redundant with "The study involved 123 people with MS who were observed to have gastrointestinal and psychological problems" in the previous section.
Therefore, it is better to simply state Exclusion criteria:
2.7. Exclusion criteria
Exclusion criteria were advanced periodontal disease, colorectal neoplasia, and
patients undergoing psychiatric antidepressant treatment.
Results section 3.1 states "10 specific batteries from crevicular fluid were evaluated...." I assume that this is a typo and "batteries" should read bacteria.
However, the Results then states "From a bacteriological point of view, 11 bacteria from the gingival crevicular fluid were tested...."
1. Are the authors making a distinction between crevicular fluid and gingival crevicular fluid?
2. Were two separate test performed, one assessing the presence of the 10 bacteria groups listed in the first sentence and the second assessing the presence of these 10 bacterial groups plus Fusobacterium nucleatum?
The Results section 3.2 states "Malnutrition differs between the 3 groups insignificantly (p=0.011), and insignificant differences were recorded in 2 research groups (CG and IBS) but still insignificant between IBS and IBD (p=1.000), according to the research groups." The p value presented suggest a significance difference. This appears to be a typo. Therefore, the statement should be changed to something like
"These was no significant difference in malnutrition between the 3 groups."
The Results section 3.6 states "In the case of Capnocytophaga spp. and Actinobacillus actinomycetemcomitans bacteria, a negative value of the Pearson coefficient can be observed, which indicates an inversely proportional relationship. These 2 bacteria correlated significantly with IBS, meaning the highest presence of these bacteria correlated with the IBS group." However, Actinobacillus actinomycetemcomitans is present in more patients in the IBS group than in the CG or IBD groups, but Capnocytophaga spp. is present in fewer patients in the IBS group than in the CG or IBD groups.
Also, the most striking point about Figure X is that Prevotella intermedia is present in all of the patients in the IBD groups and in 0 patients in the CG and IBS groups.
Did the authors investigate correlation between two or more bacteria groups and the CG, IBS, IBD groups.
Again a striking correlation is that 100% of the IBD patients were infected with Eubacterium nodatum and/or Prevotella intermedia while 0 patients in the CG and IBS groups were infected with these bacteria.
The Discussion states " Among the 3 groups, the highest value of BMI was recorded in the IBS group, indicating that that it is not the number of periodontal batteries that intervene in the pro-inflammatory process but the number of infections that are relevant".
Again, I assume that "batteries" in this sentence is a typo and should be bacteria.
Also, the small difference in BMI between the groups does not support the above statement.
The Discussion states " Patients in the current study presented with visceral fat at an average level of 7.97, which is considered at risk for the body." However, the very large standard deviation suggests that in each group some patients had very large amount of visceral fat, but other patients had a low amount of visceral fat.
The Discussion states "In the current study, the IBD group recorded the highest number of periodontal bacteria present, as well as significantly more pronounced gastrointestinal symptoms." However, patients in both the IBS and IBD groups had significantly more pronounced gastrointestinal symptoms than patients in the CG group, and patients in the IBS group tended to have slightly more pronounced gastrointestinal symptoms than patients in the IBD group.
The Discussion states "The correlations of the research groups (IBS and IBD group) with the presence of bacteria is in accordance with specialized studies in the current literature." I assume that the research groups (IBS and IBD group) are the IBS and IBD groups of the present study?
The Discussion states "Another limitation was the small number of patients undergoing colonoscopy." However, there is no colonoscopy data presented in the revised manuscript.
The Conclusions state "A higher BMI was recorded for the IMS group...." I assume that IMS is a typo and should be IBS.
In all three groups, differences in malnutrition, fat mass, and visceral fat are slight but fat mass and visceral fat do correlate with malnutrition.
Table 3. Descriptive presentation of the parameters of gastrointestinal disorders in the three research groups is mislabeled. It should be Table 4. tables 4,5,6,7 also need to be relabelled.
All abbreviation need to be defined at first mention.
For numbers, decimal points should be periods "." not commas ",".
There are 3 groups in the present study; CG vs IBS, CG vs IBD, and IBS vs IBD.. All significant differences should be noted and the groups being compared should be specifically stated.
Author Response
Primarily, we, the authors of the present manuscript wish to thank you for thoughtful commentary you have provided to improve the quality of the paper. We are very grateful for the time and effort you have devoted to this task. We have extensively revised our manuscript according to the recommendations. All changes in the text and the new figures that we have redesigned are highlighted. Please, see the point-by-point answers to your comments below.

This manuscript is a resubmission of an earlier submission. The following is a list of the peer review reports and author responses from that submission.
Round 1
Reviewer 1 Report
The manuscript by Timea Claudia Ghitea states that "The existence of a direct connection between periodontal bacteria and intestinal inflammation, which is correlated with colorectal cancer, was concluded." However, there is very little information presented regarding intestinal inflammation or colorectal cancer. Figure 3A presents colonoscopy data from the 3 groups of study participants. 5 patients in the control group, 4 patients in the group with periodontal Fusobacterium nucleatum, and 5 patients without periodontal Fusobacterium nucleatum but with at least one other type of periodontal bacteria had intestinal inflammation. Thus, there is no connection between periodontal bacteria and intestinal inflammation. Figure 3B presents another set of colonoscopy data from the 3 groups of study participants. 3 patients in the control group, 4 patients in the group with periodontal Fusobacterium nucleatum, and 3 patients without periodontal Fusobacterium nucleatum but with at least one other type of periodontal bacteria had colorectal tumors. Thus, there is no connection between periodontal bacteria and colorectal tumors. The manuscript states that a significantly higher incidence of tumor formation was found in the FG group (p=0.001). However, the data showing a higher incidence of tumor formation in the FG group with a significance of p = 0.001 is not presented in the manuscript. However, strains of bacteria are becoming known as possibly oncogenic. In addition to Helicobacter pylori, which is a group 1 human carcinogen, Fusobacterium nucleatum has been shown to potentiate intestinal tumorigenesis in mice. Therefore, this study reports valuable human data.
The manuscript needs to be rewritten and the author needs to avoid unwarranted conclusions. In addition, the title should be changed as the results do not show a correlation between periodontopathies and colorectal cancer.
Additional Comments:
Section 2.6 in Methods states "From the cohort, 81 people performed the periodontopathogenic test (Micro-IDent)". However, the periodontopathogenic test should have been performed on all 123 patients in the study.
Section 3 (Results) states that 52 study participants were from urban environments and 71 from rural environments (t = 35.266, p = 0.001). This suggests that the student t-test was used. However, t-tests are used to compare means and there are no means in the demographic description of the study participants.
The Results then states "with an average age of 40.27 years. (SD 15.35) years (t = 29.099, p = 0.001)". There is no comparison of means of different groups being made here, the text is simply stating the average of the study participants. Therefore, the is no t value or p value.
In Table 1, the % column is comparing the participants of the individual groups to the total study population. Instead, the % column should compare the participants of the individual groups to the group population. For example, the % of participants in the CG group from an urban environment is 13/42 = 31.0%.
The Results state "Baseline anthropometric parameters were thus cohort mean BMI 30.85 (SD 7.69) t = 44.500 p = 0.001". However, there does not appear to be a difference in BMI values between groups. In addition, when describing the results presented in Table 2, which groups are being compared must be explicitly stated: FG vs CG, ABG vs CG, and FG vs ABG.
The Results state "Thus, it can be seen that periodontal infections occur at young ages (34.05±9.40 years), compared to the control group (44.55±15.80 for years)." However, the FG group also has periodontal infections. Therefore, the sentence needs to be rewritten.
In Figure 2, only the results for group ABG should be shown as by definition all 41 study participants in the FG group have Fusobacterium nucleatum in the crevicular fluid. Also, in this figure there should be no error bars.
Section 3.2 states "Evaluating gastrointestinal problems, nausea/vomiting, bloating, diarrhea, constipation and malnutrition were observed in the 3 study groups, shown in table 3." The words used in the text and the words used in the Table should be the same. If "bloating" = "Flatulence" then the same term should be used in the text and Table 3.
Section 3.2 states "Moderate malnutrition was mostly observed in the control group, which can be correlated with constipation. On the other hand, in the group with Fusobacterium infection, moderate malnutrition was registered with an increase of 3.2%, affecting 80.5% of patients in total." This suggests that there was an increase in moderate malnutrition between the CG and FG groups. As shown in Table 3 there is no difference in moderate malnutrition between the CG and FG groups.
In Tables 3, 4, and 5 (as I noted for Table 1) % should refer to the percent of the study participants in the group rather than percent of the total number of study participants.
Section 3.4 states "Tumor formations had a 20% higher incidence in the FG group than in the other two groups (Figure 3B)." This suggests that the incidence of tumors in the FG group was greater than in the other two groups. It is not correct to base such a conclusion on a single individual.
Also, Fig. 3B is confusing. The Methods state that 5 participants from each group underwent colonoscopy. Why are only 4 patients from each group shown in Fig. 3B?
Section 3.6 states "Pearson correlation shows a strong relationship between smoking and colorectal cancer risk of 78.3% in patients with a higher incidence in the FG group." Risk can not be concluded using only 15 of 123 study participants. The actual data must be shown. Of the (presumably) 15 participants who underwent colonoscopy which participants were smokers and which participants had tumors?
Section 3.6 states "The estimate of colorectal cancer risk in the three study groups (represented by R2) shows an incidence of 92.0% among drinkers in the ABG group. (Table 6 and Figure 4). The ANOVA test demonstrates statistical significance." Given the small number of study participants who underwent colonoscopy, the above statement is misleading. Statistical significance can not be concluded using only 15 of 123 study participants. The actual data must be shown. Of the (presumably) 15 participants who underwent colonoscopy which participants drank alcohol and which participants had tumors?
In section Fig. 5, EORTC-QLQ CR29, VAS and WCRF/AIRC values for the individual CG, FG, and ABG groups should also be shown.
Author Response
Firstly, we, the authors of the present manuscript wish to thank you for thoughtful commentary you have provided to improve the quality of the paper. We are very grateful for the time and effort you have devoted to this task. We have extensively revised my manuscript according to the recommendations. All changes in the text and the new figures that we have redesigned are highlighted. Please, see the point-by-point answers to your comments below. All correction was highlighted in the manuscript.

Reviewer 2 Report
This study focus on the correlation between the colorectal tumor diseases and the presence of Fusobacterium nucleatum in the gingival crevicular fluid. With this study, the existence of a direct connection between periodontal bacteria and intestinal inflammation, which is correlated with colorectal cancer, was concluded. This is a very interesting study and can be an important enlightenment for the readers of this journal. However, the English language and style of this manuscript should be improved, i.e. in Abstract, the colon after the second sentence is used irregularly and the words “pres-ence” and “signifi-cantly” should be corrected.
Author Response
Firstly, we, the authors of the present manuscript wish to thank you for thoughtful commentary you have provided to improve the quality of the paper. We are very grateful for the time and effort you have devoted to this task. We have extensively revised my manuscript according to the recommendations. All changes in the text and the new figures that we have redesigned are highlighted. Please, see the correction highlighted in the manuscript.

Reviewer 3 Report
Fusobacterium nucleatum (FG), which is analyzed to be mainly correlated with colorectal cancer in this manuscript, is a Gram negative, anaerobic oral bacterium that plays a major role in forming dental plaques in the human oral cavity. In health and disease, FG is a key component of dental plaque due to its abundance and its ability to coaggregate with other bacteria species in the oral cavity. In most cases, FG infection exists in people with dental plaque. Therefore, it is difficult to analyze that there is a correlation between periodontal disease and colon cancer only with the presence of FG.
In order to analyze the correlation between periodontal disease and colorectal cancer, it is recommended to perform a dental examination that analyzes the health and inflammatory condition of periodontal tissue.
Author Response
Firstly, we, the authors of the present manuscript wish to thank you for thoughtful commentary you have provided to improve the quality of the paper. we are very grateful for the time and effort you have devoted to this task. We have extensively revised my manuscript according to the recommendations.

Round 2
Reviewer 1 Report
There remain several changes that need to be made to the manuscript by Lavinia Alina Rat et al.
1. "(1)", "(2)", "(3)", and "(4)" should be removed from the abstract.
2. The sentence "The study was carried out on 123 people, which were divided into approximately 3 equal groups, randomized: the control group (CG) of 42 people, a group that had Fusobacterium nucleatum (FG) infection 41 people and a group that had gingival infections with one of the 10 bacteria tested (ABG) of 40 people, in which the presence of gastrointestinal and psychological problems were observed."
should be changed to
"The study was carried out with 123 people in whom the presence of gastrointestinal and psychological problems were observed. Participants were divided into approximately 3 equal groups. The control group (CG) was composed of 42 people who did not have gingival infections. Group 2 was composed of 41 participants who had Fusobacterium nucleatum (FG) infection. Group 3 was composed of 40 participants who had gingival infections with at least one of the bacteria identified by the micro-IDent test (ABG) other than Fusobacterium nucleatum."
3. The Abstract states "A significantly higher incidence of tumor formations were found in the FG group (p=0.001)".
The study report the results of colonoscopy on 15 of the study participants.
3 participants in the CG group had colon tumors.
4 participants in the FG group has colon tumors.
3 participants in the ABG group had colon tumors.
Therefore, there was no statistical difference in colon tumor incidence between the groups.
Therefore, "A significantly higher incidence of tumor formations were found in the FG group (p=0.001)" must be removed from the manuscript.
The sentence "A colonoscopy was performed on 5 people from each group, who presented the most intense symptoms; (3) A significantly higher incidence of tumor formations were found in the FG group (p=0.001), and the non-specific intestinal inflammation was present in the 3 groups with insignificant differences."
should be replaced with something like
"All study participants underwent anthropometric and quality of life testing. A colonoscopy was performed on five participants from each group who presented the most intense symptoms. There were no significant differences in intestinal inflammation or tumor formation between the 5 participants from each group who had a colonoscopy. At the beginning of the study, the group in which the quality of life was most affected, according to the EORTC QLQ – CR29 test, was the FG group (p=0.023)."
4. The Abstract states that "The significantly higher presence of multiple gastrointestinal problems together with periodontal problems with the presence of Fusobacterium nucleatum bacteria...."
The study reports the following differences between the GC and FG groups:
Absent Present
Nausea/vomiting (CG) 24 18
Nausea/vomiting (FG) 19 22
No difference between CG and FG
Flatulence (CG) 26 16
Flatulence (FG) 12 29
Significantly higher in the FG group
Diarrhea (CG) 15 27
Diarrhea (FG) 13 28
No difference between CG and FG
Constipation (CG) 0 42
Constipation (FG) 20 21
Significantly higher in the CG group
Absent Moderate Severe
Malnutrition (CG) 7 35 0
Malnutrition (FG) 7 34 0
No difference between CG and FG
Therefore, there is no overall difference in gastro-intestinal problems between the CG and FG groups.
Therefore, the statement "The significantly higher presence of multiple gastrointestinal problems together with periodontal problems with the presence of Fusobacterium nucleatum bacteria, as well as the presence of intestinal inflammation in the CG, justifies that this correlation can be an important indicator of serious colorectal diseases" must be removed from the manuscript.
5. I assume that the health/colorectal disease of the study participants will continue to be followed by the authors. If that is correct, then the concluding sentences in the Abstract can be something like
"While the present study did not find a statistically significant correlation between periodontopathic bacteria and the severity of colorectal disease, it is notable that 4 of the 5 participants in the FG group who underwent a colonoscopy did have colorectal tumors. Follow up of the study participants will help to determine whether periodontopathies can be used as indicators of severe colorectal disease. In addition, this study needs to be extended to a larger group of patients."
Also, if the health/colorectal disease of the study participants will continue to be followed by the authors, the title of the manuscript can be changed to something like
Can the correlation of periodontopathies with gastrointestinal diseases be used as indicators in severe colorectal diseases: A preliminary report.
6. The final paragraph of the Introduction states "The aim of the paper is to establish a direct link between gastrointestinal symptoms and the presence of the bacterium Fusobacterium nucleatum with a view to early diagnosis or prevention of the development of colorectal cancer in patients at risk."
It is not correct to state that the aim of the study was to establish a direct link... Rather, the Introduction should conclude with
"Fusobacterium has been identified as a potential oncogenic bacteria [https://doi.org/10.1146/annurev-immunol-051116-052133]. The present report is an initial report of our investigation into whether or not there is a link between the oral presence of the bacterium Fusobacterium nucleatum and symptoms of gastrointestinal disease, and if such a link exists to determine if it can be exploited for early diagnosis of colorectal cancer or prevention of the development of colorectal cancer in patients at risk."
7. Section 2.1.1 in Methods states "The variations in the four independent groups...." However, the study was composed of only three groups: CG, FG, and ABG.
8. Section 2.6 states "The study involved 123 people with MS, divided into 3 approximately equal groups, randomized, with the help of the website random.org [24]:"
This statement is not correct. Randomization occurs before diagnosis. In the present study, the participants in the FG and ABG groups were divided into these groups depending on the results of the micro-IDent test.
Also, this section does not define the Control Group. For this manuscript to be acceptable for publication, it is essential that the Control Group be specifically defined. If the oral status of the control group (or other factors that distinguish the Control Group from the FG and ABG groups) is unknown, then comparison of the control group with the FG and ABG groups is meaningless and the manuscript should be withdrawn.
Also, the 10 bacteria in the gingival crevicular fluid ABG group should be defined.
Finally, the health status and quality of life evaluations is presented in the Results, not the Methods.
Therefore, if the participants in the CG group can be distinguished from the participants in the FG and ABG groups (presumably by oral status) Section 2.6 needs to be rewritten. For example:
"2.6. Study groups
The study involved 123 people with MS. Dental examination identified 42 participants as not having oral infection. These participants constituted the Control Group (CG). A micro-IDent test was performed on the 81 other participants. 41 participants had oral infection by Fusobacterium nucleatum and constituted the FG group. 40 participants had oral infection by one of the other 10 bacteria identified by the micro-IDent test and constituted the ABG group."
9. Section 2.7 states that the inclusion criteria were patients with gastrointestinal symptoms and periodontal problems. However, the authors previous response to my comment #8 was "The patients come from the medical clinic of clinical nutrition, and they presented with gastrointestinal problems. This is the starting point of the study, therefore all patients included in the study presented with gastrointestinal problems." Also, the abstract defines the study participants as having gastrointestinal and psychological problems, and no mention of periodontal problems.
Therefore, "and periodontal problems" should be removed.
10. The Results state
"Thus, periodontal infections tested by Micro-IDent®plus tests were confirmed even at young ages (34.05±9.40 years) in the 2 study groups, compared to the control group (44.55±15.80 years)."
and
"From a bacteriological point of view, 11 bacteria from the gingival crevicular fluid were tested, and the presence of Fusobacterium nucleatum was followed (figure 2). It was observed to be most frequently present in the crevicular fluid in 33.3% of the total study subjects."
suggesting that all 123 study participants had mcro-IDent tests.
However, Section 2.6 in Methods and the answer to my comment #8 indicates that the participants in the CG group did not have a micro-IDent test.
The micro-IDent testing status of the CG group needs to be stated.
If the participants in the CG group did not have a micro-IDent test, then the CG group can be defined as "Dental examination identified 42 participants as not having oral infection. These participants constituted the Control Group (CG)."
If the participants in the CG group did have a micro-IDent test, then the CG group can be defined as "Micro-IDent testing results were negative for 42 study participants. These participants constituted the Control Group (CG)."
11. Section 3.2 states "Malnutrition differs between the 3 control groups insignificantly (p=0.187), significant differences were only recorded in the case of moderate and severe malnutrition in 2 research groups (FG and ABG), verified by the Chi-square technique (X2=17.780 p=0.001) according to research groups."
The results presented in Table 3 are
Absent Moderate Severe
Malnutrition (FG) 7 34 0
Malnutrition (ABG) 8 30 2
Comparing moderate and severe malnutrition in the FG and ABG groups using a 2x2 contingency table and the Chi-square test (GraphPad) yields a two-tailed P value of 0.1388.
12. Section 3.2 states "On the other hand, in the group with Fusobacterium infection, moderate malnutrition was registered with an increase of 3.2%, affecting 80.5% of patients in total." Why is a very slight increase in the percent of participants in the FG group (82.9) compared to the total study participants (80.5) important? If it is not important, the sentence should be deleted.
13. In Section 3.4, the results of the colonoscopic examination of all 15 patients needs to be shown. Basing Fig. 3B on only the positive result from group FG can be seen as data manipulation.
14. The Discussion states "In the current study, the crevicular fluid analyzes showed a high incidence of Fusobacterium nucleatum, which further led to the verification of the gastrointestinal status by clinical examination (nausea/vomiting, flatulence, diarrhea, constipation, malnutrition) and by colonoscopy. The results of the clinical examination show that 48.0% of patients presented with retching/vomiting, 56.9% with flatulence, 70.0% with diarrhea, 60.2% with constipation and 82.1% with moderate and severe malnutrition." This passage is misleading. The patients were initially selected because they had gastrointestinal problems. If the general population was tested, it is very possible the people with oral Fusobacterium nucleatum would have far lower nausea/vomiting, flatulence, diarrhea, constipation, and malnutrition.
Also, the percentages noted are from the total study population, not the FG group.
Therefore, this passage should simply state
"In the current study, the crevicular fluid analyzes showed a high incidence of Fusobacterium nucleatum in patients with gastrointestinal problems."
15. The next paragraph in the Discussion "After assessing the psychological factors...." discusses findings from the total study population. Therefore, it should be moved up so that it is paragraph #2 in the Discussion. This keeps the discussion of the general population together and the Discussion of oral bacteria and intestinal problems together.
16. The Discussion states "Our results show that the most affected in terms of quality of life are those from the FG group, where a significantly higher incidence of colorectal tumor diseases were recorded". However, a significantly higher incidence of colorectal tumor diseases was not recorded.
Therefore, the passage "where a significantly higher incidence of colorectal tumor diseases were recorded, which is also associated with intense pain, but of which does not affect any daily activities" must be deleted from the manuscript.
17. Also, in the sentence "Failure to comply with these together with general risk factors, lead to health impairment, intestinal health imbalance, which if associated with Fusobacterium nucleatum infection, increases the risk of colorectal cancer" the passage "which if associated with Fusobacterium nucleatum infection, increases the risk of colorectal cancer" must be deleted from the manuscript.
18. Also, the sentence "Therefore, in patients with specific gastrointestinal symptoms, and an altered psychological state, it would be indicated to check the periodontal crevicular fluid, in order to be able to intervene either in case of
early-stage cancer, or without tumor formation, only with lifestyle changes" needs to be rewritten. For example:
"Therefore, in patients with specific gastrointestinal symptoms and an altered psychological state, lifestyle changes are indicated."
19. The final sentence in the Discussion stated "Another limitation may have been the small number of patients undergoing colonoscopy."
This should be rewritten
"Another limitation was the small number of patients undergoing colonoscopy."
20. The Conclusion can be rewritten:
5. Conclusions
A higher BMI was recorded for the ABG group, but the increase was not statistically significant. This may suggest a more complex link with periodontal bacteria, not just Fusobacterium. The highest fat mass was observed in the FG group, which can be explained by the pro-inflammatory process of the digestive tube, and is a high risk for colorectal cancer. Visceral fat was highest in the CG group, which explains the insignificant link of visceral fat with the presence of bacteria.
Among the 11 bacteria present in the crevicular fluid detectable by the micro-IDent test, Fusobacterium nucleatum was detected most often. The presence of serious gastrointestinal problems was recorded in the FG group, which was the basis for colonoscopy recommendations. The quality of life most affected, according to the EORTC QLQ – CR29 test, was in the FG group (p=0.023).
Pain pre-sent, but not affecting daily activities, evaluated by VAS test, was observed in all 3 groups. In the case of the WCRF/AICR test, regarding compliance with the specific recommendations, a partial compliance with the recommendations was observed, the groups with the highest compliance was the ABG group. Failure to comply with the specific recommendations can lead to worsening of the health condition, and affect the quality of life.
Fusobacterium has been identified as a potential oncogenic bacteria [https://doi.org/10.1146/annurev-immunol-051116-052133]. The present report is an initial report of our investigation into whether or not there is a link between the presence of the bacterium Fusobacterium nucleatum in gingival crevicular fluid and symptoms of gastrointestinal disease, and if such a link exists to determine if it can be exploited for early diagnosis of colorectal cancer or prevention of the development of colorectal cancer in patients at risk. Inflammation in the colon and development of colorectal tumors was observed in all three groups. Follow-up of the study participants will help to determine if periodontopathies can be used as indicators of severe colorectal disease."
Reviewer 3 Report
I read the revised manuscript very carefully.
I don't think there was any correction in the part that I recommended to revise.
In my opinion, it is difficult to analyze the relationship between colorectal cancer and periodontopathies just by the presence of Fusobacterium nucleatum. In addition to the presence of Fusobacterium nucleatum, there are various diagnostic methods and indexes to diagnose the condition of the periodontal tissue to diagnose periodontal disease. After evaluating the index of periodontal disease by diagnosing it in detail like this, the analysis between them and the colorectal cancer will be possible to analyze the correlation between periodontopathies and the colorectal cancer as the title of this manuscript.
Author Response
We, the authors of the present manuscript wish to thank you for the thoughtful commentary you have provided to improve the quality of the paper. We are very grateful for the time and effort you have devoted to this task. We have extensively revised our manuscript according to the recommendations. The methods were completed with section 2.2. that we have redesigned and highlighted.
